# Poly(methyl methacrylate) Nanocomposite Foams Reinforced with Carbon and Inorganic Nanoparticles—State-of-the-Art

Ayesha Kausar [1],* and Patrizia Bocchetta [2]

1   Nanosciences Division, National Center for Physics, Quaid-i-Azam University Campus, Islamabad 44000, Pakistan
2   Department of Innovation Engineering, University of Salento, Edificio La Stecca, via per Monteroni, 73100 Lecce, Italy; patrizia.bocchetta@unisalento.it
*   Correspondence: asheesgreat@yahoo.com

**Abstract:** Polymeric nanocomposite foams have attracted increasing research attention for technical reasons. Poly(methyl methacrylate) is a remarkable and viable thermoplastic polymer. This review highlights some indispensable aspects of poly(methyl methacrylate) nanocomposite foams with nanocarbon nanofillers (carbon nanotube, graphene, etc.) and inorganic nanoparticles (nanoclay, polyhedral oligomeric silsesquioxane, silica, etc.). The design and physical properties of poly(methyl methacrylate) nanocomposite foams have been deliberated. It has been observed that processing strategies, nanofiller dispersion, and interfacial interactions in poly(methyl methacrylate)–nanofiller have been found essential to produce high-performance nanocellular foams. The emergent application areas of the poly(methyl methacrylate) nanocomposite foams are electromagnetic interference shielding, sensors, and supercapacitors.

**Keywords:** poly(methyl methacrylate); nanocomposite; foam; electromagnetic; sensor; supercapacitor

## 1. Introduction

Poly(methyl methacrylate) (PMMA) is a commercially important thermoplastic polymer with appropriate chemical resistance, corrosion resistance, and anti-weathering properties toward the methodological applications [1–3]. However, the thermal and mechanical stability properties of PMMA are not high enough to meet industrial demands. Consequently, various design modifications have been carried out on this polymer to enhance its physical properties. The PMMA matrix has been reinforced with nanoparticles, to develop nanocomposites. Consequently, the technical application areas of PMMA-based nanocomposites have been focused on the fields of materials science and nanotechnology. PMMA has been used to form a three-dimensional cellular foam structure [4]. PMMA foam possesses inherently high thermal, mechanical, optical, sensing, and environmental properties, relative to neat PMMA. In this regard, nanocarbon nanoparticles have been employed to enhance PMMA foam matrix properties. Research has turned towards the incorporation of nanoparticles such as graphene [5], carbon nanotubes [6], nanoclay [7], inorganic nanoparticles [8], etc. in the PMMA matrix [9–11]. Consequently, high performance PMMA nanocomposite foams have been developed. The superior flexibility, thermal stability, mechanical robustness, electrical conductivity, sensing, capacitance, and radiation shielding properties of PMMA and nanofiller-based nanocomposite foams are appropriate for several technical applications. In this review, progress in the design, features, and applications of PMMA nanocomposite foams has been offered. Advanced PMMA nanocomposite foams have been reproduced in several wide-ranging as well as promising application areas. The future of PMMA nanocomposite foams relies on the design of modified nanoparticle-based PMMA aerogels.

## 2. Poly(methyl methacrylate)

Poly(methyl methacrylate) (PMMA) is a transparent thermoplastic polymer [12,13]. It is made up of methyl methacrylate monomer. It was originally discovered in 1930s [14]. It is a lightweight polymer with a density of 1.2 $gcm^{-3}$. PMMA shows atacticity, isotacticity, and syndiotacticity in its structure. PMMA is an optically transparent polymer and has been frequently used as inorganic glass [15]. PMMA has a refractive index of 1.49 [16–18]. PMMA has an amorphous nature, chemical resistance, weather defiance, and corrosion resistance properties. The thermal stability of PMMA has been extensively deliberated [19]. PMMA has a glass transition temperature in the range of 100–130 °C. Using the methyl methacrylate monomer, the solution, bulk, suspension, emulsion, free radical, atom transfer radical, and anionic radical chain polymerization methods have been used to form the PMMA backbone [20–23]. However, neat PMMA does not possess enough thermal/mechanical stability to meet a range of technical demands [24,25]. In this regard, high performance PMMA-based nanocomposites have been reported [26,27]. Various nanofillers employed within the PMMA matrix are graphene [28], carbon nanotube (CNT) [29], fullerene [30], layered silicate [31], silica [32], alumina, polyhedral oligomeric silsesquioxane [33], and metal nanoparticles [34,35]. PMMA has been applied in numerous applications including automotive parts, coatings, additives, neutron stoppers, packaging, and the nanocomposite industry.

## 3. Poly(methyl methacrylate) Foam

Polymeric nanocellular foams have been produced using various processes [36,37]. Among the most promising foaming techniques are supercritical carbon dioxide ($CO_2$) dissolution, the high pressure method, and the use of foaming agents [38,39]. The plasticization effect of foaming may influence the thermal stability, polymer glass transition temperature, density, and mechanical properties [40]. Thin polymer films can be simply foamed using $CO_2$ gas dissolution [41,42]. Consequently, foams with medium-to-low density have been obtained. PMMA has been developed to form foam structures having low density, fine toughness, rigidity, and thermal conductivity properties [43]. PMMA foams have been produced using various polymeric methods [44,45]. Pinto et al. [46] adopted the $CO_2$ gas foaming process to form nanocellular and microcellular PMMA foams. The influence of $CO_2$ saturation temperature on PMMA foaming was explored. Figure 1 represents scanning electron microscopy (SEM) images showing the effect of saturation temperature on the cellular structure. The $CO_2$ saturation temperature seems to impact the foaming mechanism via better nucleation and cell growth. Zhou et.al. [47] developed PMMA microporous foam structures via hot melt pressing. The melt method was assisted by the supercritical $CO_2$ foaming method. Figure 2 shows the fabrication process for the PMMA foams. The PMMA was converted into sheets using melt hot pressing at 200 °C (40 MPa). The PMMA sheet thickness was adjusted in the range of 0.45–1.5 mm. Primarily, single-layer PMMA sheet, 25-layer PMMA sheet, and 80-layer PMMA sheet were studied.

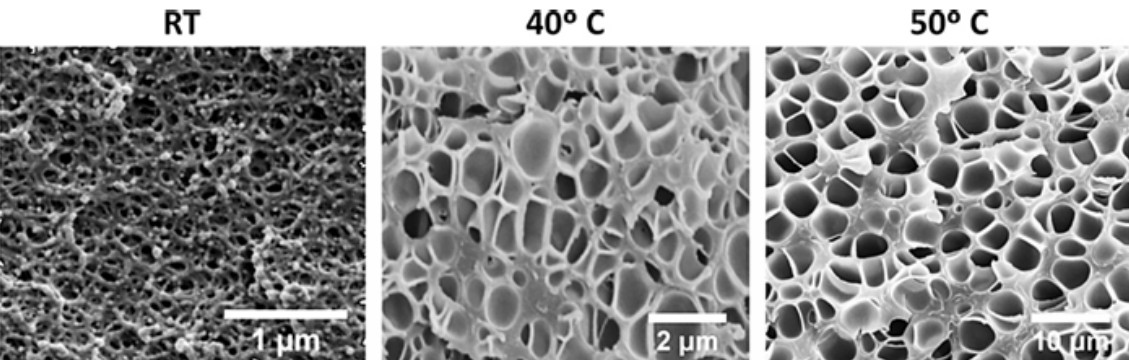

**Figure 1.** Cellular structure of PMMA foams produced after $CO_2$ saturation (30 MPa) at room temperature (RT), 40 °C, and 70 °C [46]. Reproduced with permission from Elsevier.

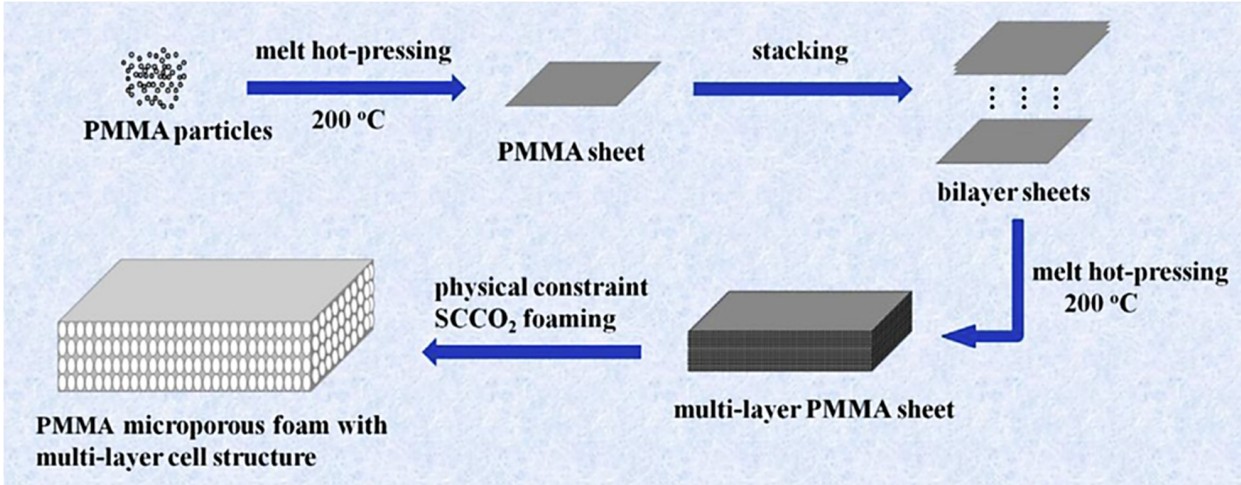

**Figure 2.** Schematic diagram of preparing PMMA microporous foam with multi-layer cell structure [47]. SCCO$_2$ = supercritical carbon dioxide. Reproduced with permission from Elsevier.

The volume density of the foams was found to decrease with rising temperatures (Table 1). This decrease in the volume density was probably due to the higher cell density of the foam at higher temperature. The compressive strength of the multi-layer foam was increased from 11.84 MPa to 20.27 MPa with the increasing PMMA layers in the structure (Figure 3). The highest compressive strength was obtained with the 80 multi-layer PMMA sheet. Inclusion of the multi-layer PMMA structure promoted better nucleation and growth of the cells in the polymer matrix [48]. Consequently, the compressive strength of the foams was increased with layering. The multi-layered foaming method has been well established for PMMA in literature [49].

**Table 1.** The volume density of the foams foaming at 16 MPa and different temperatures [47]. Reproduced with permission from Elsevier.

| Temperature (°C) | Volume Density/Psi (gcm$^3$) |
|---|---|
| 80 | $0.355 \pm 1.8 \times 10^{-3}$ |
| 110 | $0.265 \pm 2.1 \times 10^{-3}$ |
| 140 | $0.096 \pm 3.8 \times 10^{-4}$ |

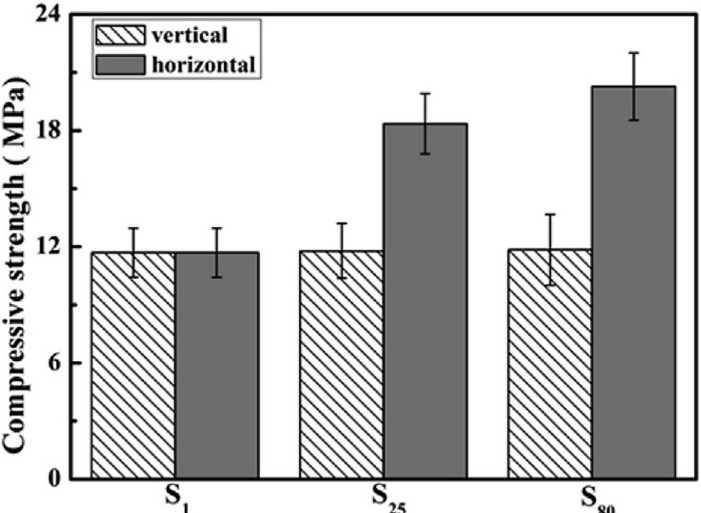

**Figure 3.** Compressive strength of foams (foaming at 16 MPa and 80 °C) [47]. The single layer PMMA sheet, 25 multi-layer PMMA sheet, and 80 multi-layer PMMA sheet were named as S1, S25, and S80, respectively. Reproduced with permission from Elsevier.

## 4. Poly(methyl methacrylate) Nanocomposite Foam

### 4.1. Poly(methyl methacrylate)/Nanocarbon Nanocomposite Foam

The carbon nanotube is an important type of nanocarbon nanoparticle with extraordinary optical, electrical, magnetic, mechanical, and thermal properties [50]. The CNT has been effectively used as reinforcement in the polymeric nanocomposites [51]. Addition of CNT has been used to escalate the foam nucleation process in the polymeric foams. The nanotube has been used to alter the polymeric foam morphologies. Chen et al. [52] prepared poly(methyl methacrylate)/carbon nanotube (PMMA/CNT). The Young's modulus and collapse strength of the PMMA/CNT nanocomposite foam were enhanced by 82% and 104%, respectively. Yuan et al. [53] designed PMMA/CNT nanocomposite foams using the melt method and $CO_2$ gas foaming technique. The preparation method for the PMMA/CNT nanocomposite foam is given in Figure 4. The bilayer nanocomposite was used for hot pressing (170 °C) and subsequent foaming. The cell size and cell density of the nanocomposite foams are given in Figure 5. The cell size was decreased from 3.8 to 3.0 µm, while cell density was increased in the range of $1.8 \times 10^{10}$–$3.7 \times 10^1$ Cells/cm$^3$, by varying the CNT content from 4 to 8 wt.%. Due to the formation of a CNT network in the PMMA matrix, cell growth was restricted and cell size was decreased. Zeng et al. [54] designed PMMA/CNT using a gas foaming technique. Table 2 depicts cell size and cell density of the PMMA/CNT nanocomposite foams. The comparison has shown that the nanocomposite microcellular foams possess smaller cell size and higher cell density, relative to the neat PMMA foam.

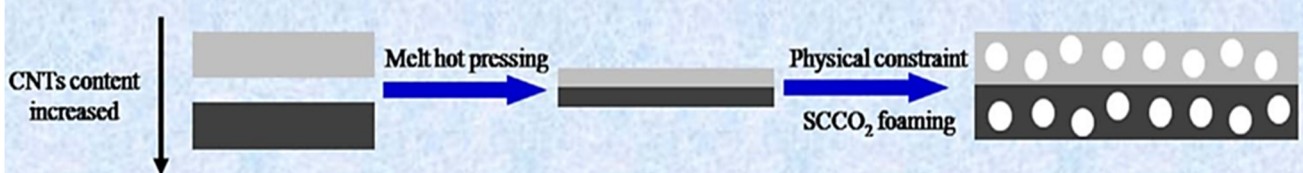

**Figure 4.** Schematic of PMMA/CNT laminated nanocomposite foam preparation [53]. PMMA/CNT = poly(methyl methacrylate)/carbon nanotube; SCCO$_2$ = supercritical carbon dioxide. Reproduced with permission from Elsevier.

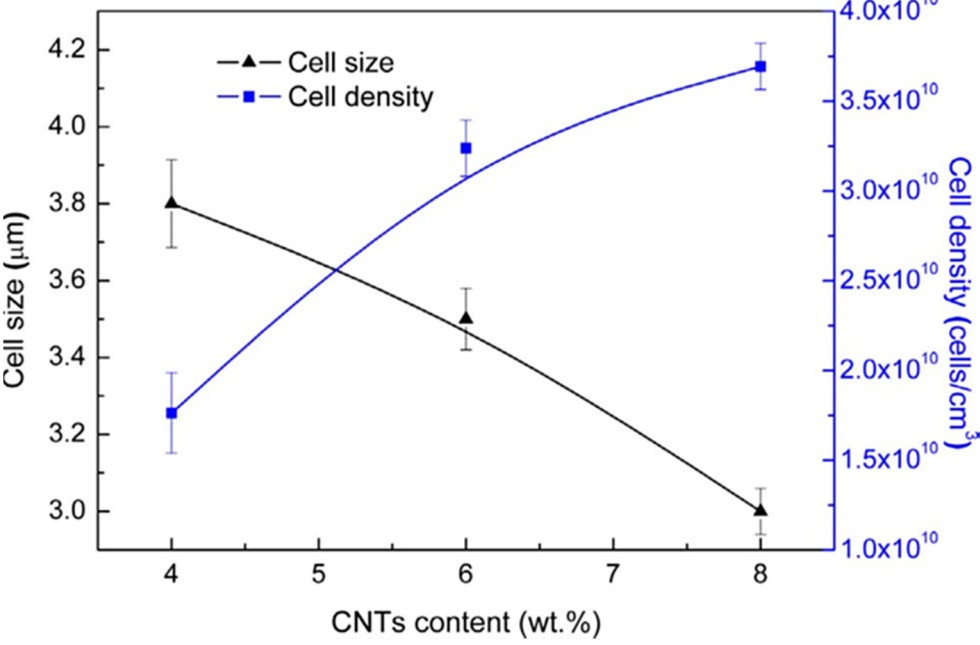

**Figure 5.** Cell size and cell density of PMMA/CNT nanocomposite foam [53]. PMMA/CNT = poly(methyl methacrylate)/carbon nanotube. Reproduced with permission from Elsevier.

**Table 2.** Comparison of cell size and cell density of PMMA and PMMA nanocomposite foams [54]. PMMA/CNT 0.5 = poly(methyl methacrylate)/carbon nanotube with 0.5 wt.% nanofiller; PMMA/CNT 1 = poly(methyl methacrylate)/carbon nanotube with 1 wt.% nanofillers. Reproduced with permission from Elsevier.

| Sample | Cell Size | Cell Density (Cell/cm$^3$) |
|---|---|---|
| Pure PMMA | $17.5 \pm 3.3$ | $1.3 \times 10^8$ |
| PMMA/CNT 0.5 | $4.2 \pm 1.0$ | $7.2 \times 10^9$ |
| PMMA/CNT 1 | $6.6 \pm 2.0$ | $1.5 \times 10^9$ |

These parameters indicate the success of the supercritical $CO_2$ process to form homogeneous nanocellular structures. Inclusion of 0.5 wt.% functional CNT enhanced the tensile strength and tensile modulus by ~40% and ~60%, compared with the neat PMMA foam. The enhancement in the mechanical properties was attributed to the increased cell density of the nanocomposite foam and better CNT dispersion. Yuan et al. [55] prepared PMMA/CNT nanocomposite foams using a supercritical foaming technique. The electrical conductivity of the nanocomposite foams was increased with the nanofiller loading from $3.34 \times 10^{-6}$ to $4.16 \times 10^{-6}$ Scm$^{-1}$ due to the CNT conductive network formation. Zeng et al. [56] proposed PMMA/CNT nanocomposite foams by a supercritical foaming method. The inclusion of nanotubes remarkably augmented cell density and reduced cell size. Addition of 1 wt.% MWCNT increased cell density and reduced cell size by 70–80 times, relative to the neat PMMA foam. Figure 6 shows the high-pressure supercritical batch foaming unit. Initially, $CO_2$ was used to feed to the pressure vessel. Then, the pressure pump was used to pass the $CO_2$ through the sample in the batch vessel for the foaming process. The temperature of the system was kept constant and monitored by temperature controller. Zakiyan et al. [57] designed a PMMA filled with CNT and graphene nanoplatelet (GNP)-based materials. They performed important studies on the dispersion and alignment of the CNT and GNP nanofillers. Surface coating of PMMA on the GNP caused hindrance in the interaction between the GNP particles in the matrix. At the same time, the CNT easily developed interconnecting network in the matrix, forming direct contacts (Figure 7). Thus, the dielectric properties of the PMMA/CNT were found to be superior, relative to the GNP nanocomposite. Graphene is a unique nanocarbon structure which has attracted immense research interest since its discovery [58]. Graphene possess superior mechanical, thermal, and electrical properties for the formation of polymeric nanocomposites [59–61]. Fan et al. [62] proposed PMMA and graphene aerogel-derived nanocomposite foams. The nanofiller used was reduced graphene oxide (rGO). The rGO was loaded in 0.67–2.50 vol.%. The electrical conductivity was increased from 0.160 to 0.859 Sm$^{-1}$. The rGO loading increased the microhardness of the nanocomposites from 303.6 to 462.5 MPa and the thermal conductivity in the range of 0.35–0.70 W/mK. The increase in the electrical and thermal properties was due to the better distribution of the nanofillers in the PMMA nanocellular structure [63,64]. Wang et al. [65] produced poly(methyl methacrylate) and graphene oxide (GO)-based PMMA/GO nanocomposite foams using the solution blending and supercritical $CO_2$ method. Figure 8 depicts SEM images of the neat PMMA and PMMA/GO nanocomposite. The inclusion of GO nanosheets increased cell density and decreased cell size. The smaller, well-defined cells indicate the homogeneous dispersion of the graphene nanosheets and fine matrix–nanofiller interaction. Both the CNT and graphene nanofillers have been used to influence the cell size, cell density, tensile strength, modulus, electrical conductivity, and thermal conductivity properties.

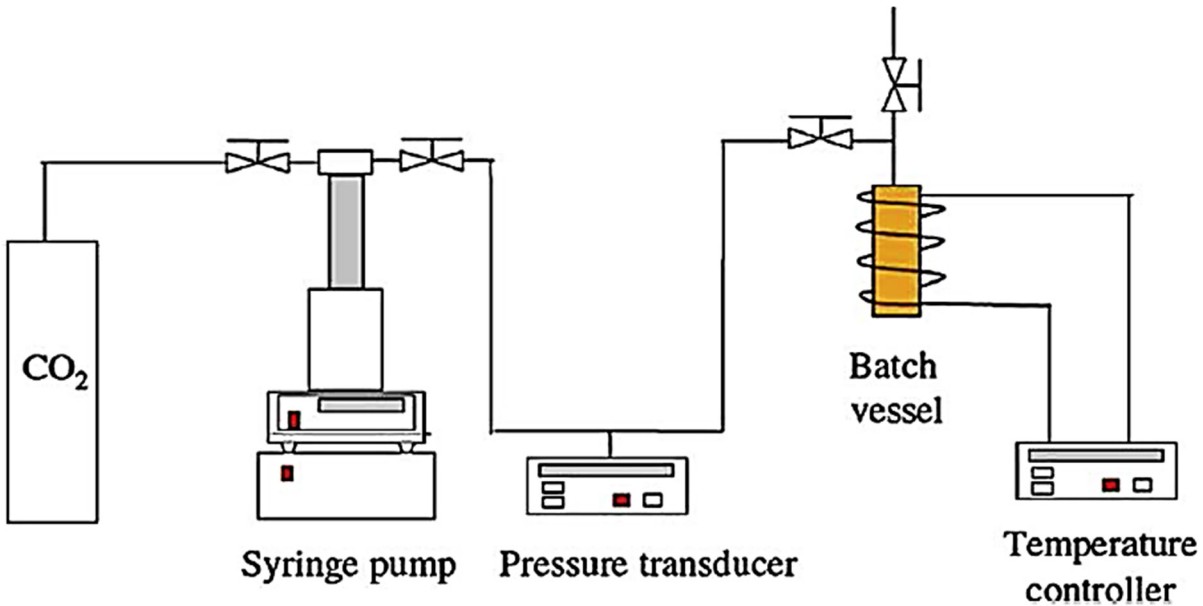

**Figure 6.** Schematic of the high pressure foaming system [56]. Reproduced with permission from Elsevier.

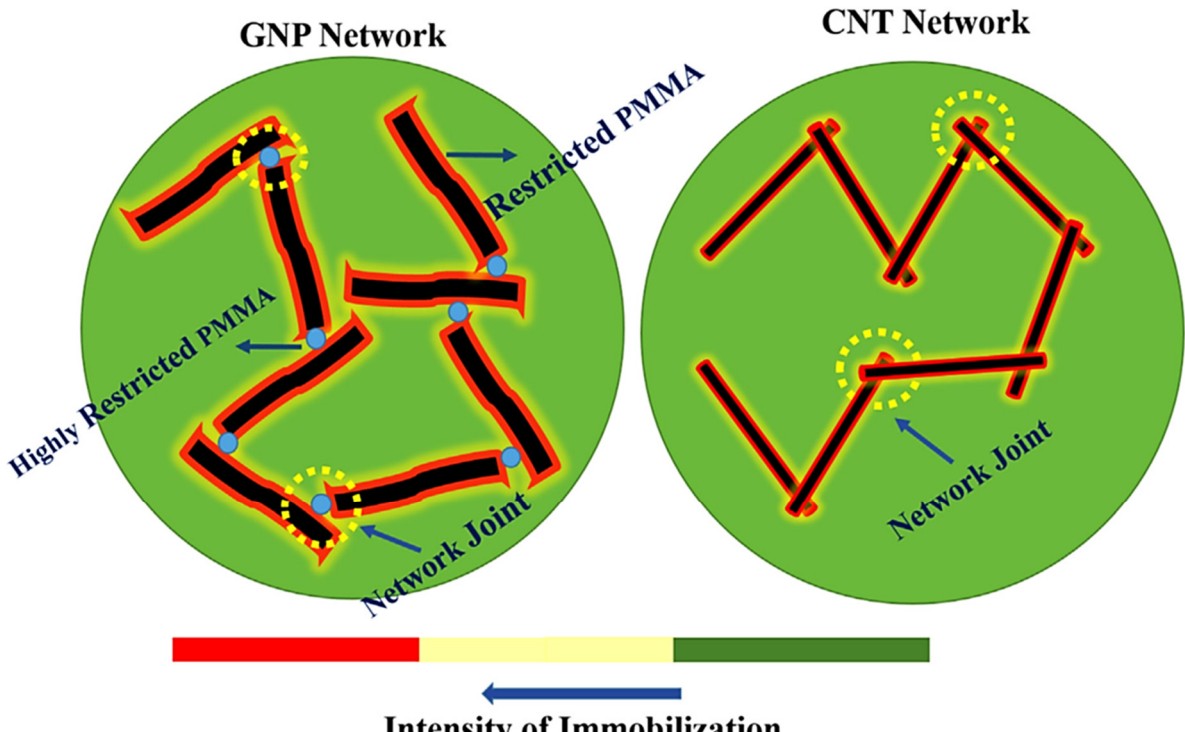

**Figure 7.** Schematic of polymer-mediated network in GNP and CNT network with direct contacts of CNT [57]. GNP = graphene nanoplatelet; CNT = carbon nanotube. Reproduced with permission from Elsevier.

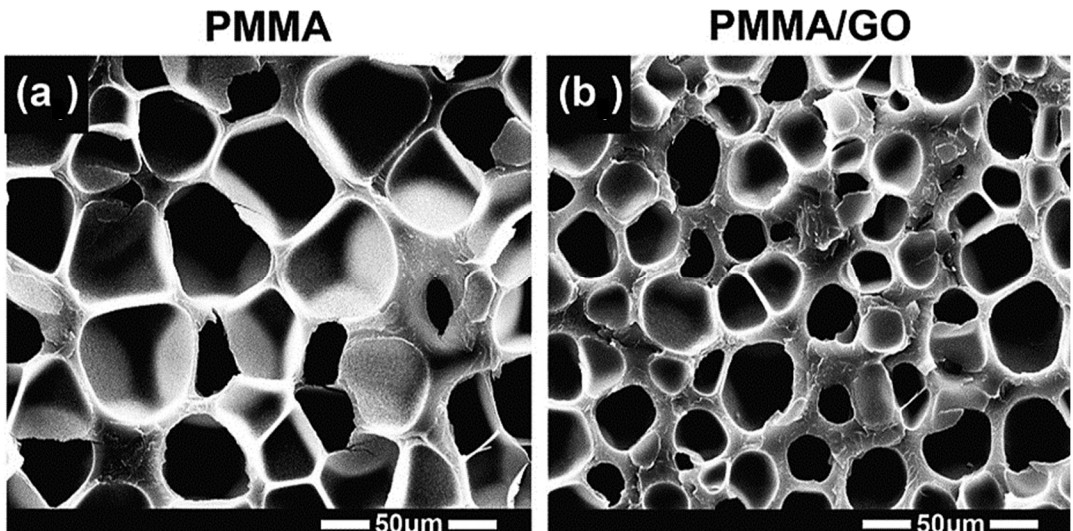

**Figure 8.** SEM images of (**a**) neat PMMA and (**b**) PMMA/GO nanocomposite [65]. PMMA/ GO = poly(methyl methacrylate)/graphene oxide. Reproduced with permission from Elsevier.

### 4.2. Poly(methyl methacrylate)/Inorganic Nanoparticle Nanocomposite Foam

Among inorganic nanoparticles, layered silicate or nanoclays have gained attention to enhance mechanical properties, thermal stability, rheology, flame retardancy, etc. of the polymer matrices [66–68]. Fu et al. [69] prepared PMMA and montmorillonite nanoclay nanocomposites and foams. The PMMA/montmorillonite unfoamed nanocomposite revealed a tensile strength of 7.74 MPa (0.5 wt.% nanoclay). The microcellular PMMA/montmorillonite foams were produced using subcritical $CO_2$ gas foaming. The 0.5 wt.% foamed sample had a tensile strength of 12.49 MPa. During the foaming process, there was the formation of homogeneously aligned nanocells and fine dispersion of the nanoparticle, resulting in enhanced mechanical properties. Realinho et al. [70] produced PMMA foams with organically modified montmorillonite in 2.5–10 wt.%. The supercritical $CO_2$ dissolution method was used. Neat PMMA had an elastic modulus of 2880 MPa.cm$^3$g$^{-1}$. Inclusion of 2.5 to 10 wt.% with organically modified montmorillonite considerably enhanced the elastic modulus from 2959 to 3449 MPa.cm$^3$g$^{-1}$. The 2.5 wt.% nanoclay revealed a glass transition temperature ($T_g$) of 140 °C using differential scanning calorimetry (DSC). The amalgamation of 10 wt.% nanoclay improved the $T_g$ of the nanocomposite foam to 142 °C. The increase in $T_g$ was due to the ordered nanoclay platelets and interaction between the PMMA and nanoclay. Ozdemir et al. [71] produced PMMA and Cloisite-based nanocomposite foam. Thermogravimetric analysis (TGA) was used to study the thermal stability of the PMMA nanocomposite foams. Neat PMMA had a maximum decomposition temperature of 374 °C. On the other hand, nanoclay enhanced the decomposition temperature to 388 °C. The increase in thermal properties was attributed to the increase in cell density, nanofiller dispersion, and PMMA domain size [72]. Moradi et al. [73] produced poly(methyl methacrylate)/polyurethane (PMMA/PU) foams reinforced with the 0.3–1.5 wt.% Cloisite nanoclay. The foaming was performed through mixing and heating techniques. According to DSC, the nanoclay loading enhanced the $T_g$ of the nanocomposite foams. Similarly, TGA thermograms showed thermal degradation resistance with the inclusion of nanoclay up to 450 °C [74]. Polyhedral oligomeric silsesquioxane (POSS) has also been used as an inorganic nanofiller in the PMMA foam matrix [75–77]. Ozkutlu et al. [78] formed PMMA and POSS-abased nanocomposite foams using a co-rotating twin-screw extruder. Neat PMMA had a thermal conductivity of 0.15 W/mK. The inclusion of 0.25 wt.% POSS raised the thermal conductivity to 0.16 W/mK. The $T_g$ of neat PMMA was 116 °C, which was raised to 119 °C with 0.25 wt.% POSS. The thermal degradation temperature of the neat PMMA (375 °C) also increased to 378 °C with nanofiller loading.

Silica has been used to form cellular foams with PMMA through the supercritical $CO_2$ foaming method, proposed by Lu et al. [79]. Here, the inclusion of 0.5 wt.% silica led to a thermal conductivity of 0.072 $Wm^{-1}K^{-1}$. The PMMA/silica foams with 5 wt.% silica enhanced the compressive strength by 92%, relative to neat PMMA. Rende et al. [80] also used supercritical $CO_2$ foaming to form the PMMA/silica foams with 0.85–3.2 wt.% silica. The cell density was increased from $7.5 \times 10^8$ to $4.8 \times 10^{11}$ cells/$cm^3$ with the nanofiller loading. Consequently, the increase in nanofiller loading reduced the average cell size by 35%. Gu et al. [81] developed a PMMA and silica aerogel using supercritical $CO_2$ foaming. The compressive strength of the PMMA/silica aerogel with 2 and 5 wt.% silica was 18.90 and 18.12 MPa, respectively. The PMMA/inorganic nanoparticle nanocomposite foams, having low density, possess high compressive strength, tensile strength, elastic modulus, $T_g$, thermal stability, and thermal conductivity properties [82,83].

## 5. Potential of Poly(methyl methacrylate) Nanocomposite Foam

### 5.1. Electromagnetic Interference Shielding

Electromagnetic absorption nanocomposites are the most in-demand materials for defense, military, and civil applications [84]. Electromagnetic absorption materials must fulfil the requirements of low density, high stability, high absorption efficiency, wide absorption bandwidth, and electromagnetic interference (EMI) shielding [85]. The polymeric nanocomposites must be filled with the electromagnetic absorbing nanofillers [86]. These nanofillers must have fine conducting/magnetic features. In this regard, carbon nanofillers such as CNT, graphene, carbon nanofiber, etc. have shown high electrical conductivity and other desired physical properties [87–89]. Chen et al. [90] studied the EMI efficiency and electrical conductivity of the polymer and graphene aerogel. Barrau et al. [91] examined the effect of the CNT nanofiller on the EMI shielding of the PMMA nanocomposite foams. Incorporation of CNT increased the electrical conductivity of the nanocomposite foam. CNT has high inherent dielectric constant, which enhanced the dielectric constant of the nanotube [92]. Das et al. [93] primed PMMA and single-walled carbon nanotube (SWCNT) nanocomposites. The SWCNT loading enhanced the percolation threshold from $10^{-15}$ to $10^{-2}$. The percolation threshold of PMMA/SWCNT was 3 wt.%. An EMI shielding of 40 dB was observed in the X-band (8–12 GHz) for 20 wt.% SWCNT at 200 MHz. Yuan et al. [53] prepared electromagnetic absorbing materials based on the PMMA/CNT nanocomposite. With a CNT loading of 4–8 wt.%, the dielectric loss of PMMA/CNT nanocomposite foams was enhanced from 2.1 to 10.8 (X-band) (Figure 9). This indicates the formation of a nanotube conducting network in the PMMA matrix with the nanofiller loading [94,95]. Consequently, the electrical conductivity of the nanocomposite foam was increased, so leading to a dielectric loss. The phenomenon of electromagnetic absorption by the nanomaterials is given in Figure 10. The laminated structure is designed to enhance the absorbing bandwidth of the PMMA/CNT nanocomposite foams. The absorbing bandwidth of the laminated nanocomposite foam was 3.5 GHz (8.9–12.4 GHz). The low dielectric constant was observed in the top layer of the foam due to lower CNT content relative to the bottom layers, which had greater nanofiller concentrations.

Thus, the lower layers revealed a high dielectric constant. Zhou et al. [96] prepared PMMA and CNT-based nanocomposite foams through a pressure foaming technique. The multi-layer cell structure was produced, having a thickness of 2 mm. Electromagnetic absorption of the nanocomposite foams was studied. The monolayer PMMA foam had an absorption bandwidth of 1.7 GHz. The PMMA/CNT foam had a higher absorption bandwidth of 2.5 GHz (9.1–11.6 GHz). Zhang et al. [97] blended PMMA with graphene and foamed it using subcritical $CO_2$. The PMMA with 1.8 vol.% graphene nanocomposite foams revealed a high electrical conductivity of 3.11 $Sm^{-1}$. The EMI shielding of the nanocomposite foams was 13–19 dB in the frequency range of 8–12 GHz. The superior properties were credited to the homogeneous dispersion of the graphene in the nanocellular foam structure.

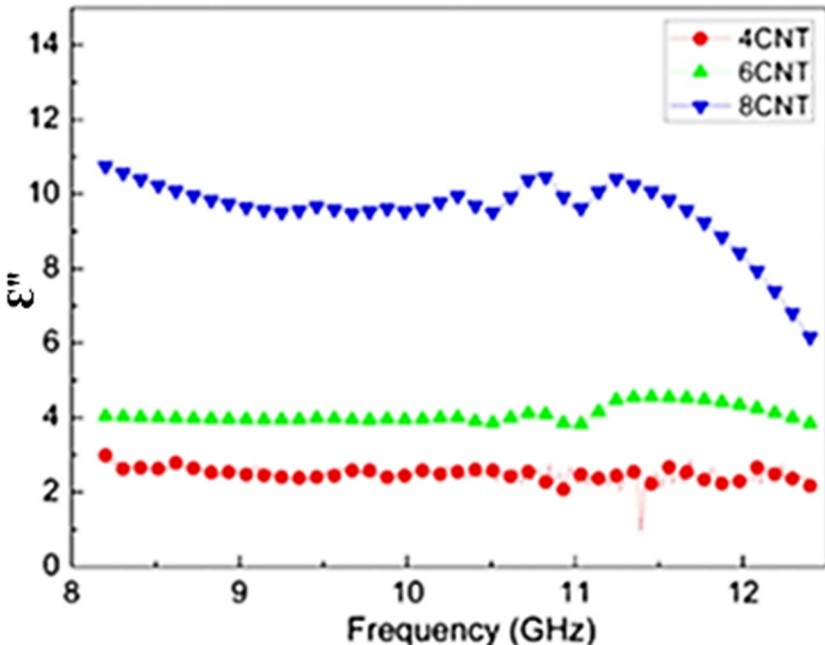

**Figure 9.** Frequency dependence of dielectric loss of PMMA/CNT nanocomposite foams [53]. Reproduced with permission from Elsevier.

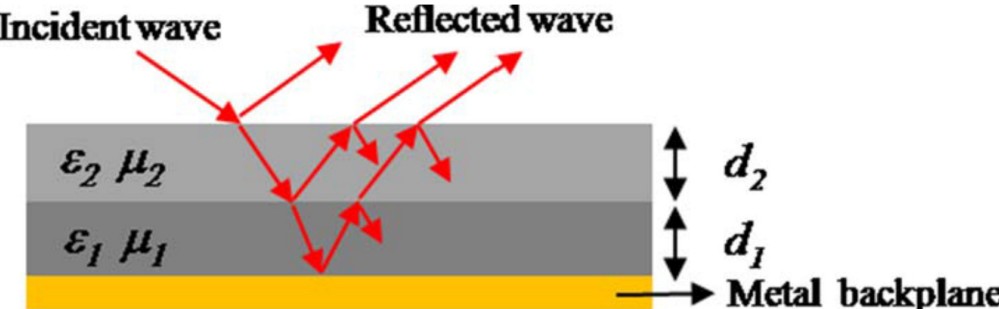

**Figure 10.** Schematic of laminated electromagnetic wave absorption materials [53]. Reproduced with permission from Elsevier.

*5.2. Sensor*

Advanced nanocomposite materials have found scope in wearable sensing devices that prevent charge accumulation caused by the metal-based materials [98]. In addition to the use of conducting polymers, the research has moved towards the use of conducting nanocomposite foams [99,100]. Use of appropriately conducting nanoparticles in the nanocomposite foam may further enhance the conductivity and sensitivity of the wearable sensor material [101]. Few PMMA-based nanocomposites have been used in wearable sensors. Vijayakumari et al. [102] proposed a conducting PMMA/Cu nanocomposite-based wearable sensor. The amount of Cu nanoparticle defined the conductivity and sensitivity of the nanocomposite sensor. The sensor may detect breathing rate, limb movement, and other physiological activities. The PMMA-based nanocomposites have also been applied in gamma ray sensors. Feizi et al. [103] formed three-dimensional PMMA and rGO-based sensing materials. The gamma ray sensor was developed by coating the PMMA/rGO nanocomposite foam on a silver-coated glass electrode. The gamma sensor had a linear response to a dose rate of 50–130 mGymin$^{-1}$. The gamma ray sensor revealed an important application in dosimetry for diagnostic activities. Few attempts have been observed on the PMMA-based nanocomposite foam sensors, but this field needs extensive research efforts to develop further.

*5.3. Supercapacitor*

A new generation of efficient energy storage devices includes the supercapacitor, ultra-capacitor, or electrochemical capacitor [104–106]. Supercapacitors usually store energy on the conducting material surface. In this regard, conducting polymers and nanocomposites have been used in advanced supercapacitors [107–109]. Few attempts have been made to produce the supercapacitors based on PMMA nanocomposite foams. Ran et al. [110] used a PMMA block copolymer-based composite as the supercapacitor electrode membrane. The electrochemical performance was established at a current density of 0.5–5 $Ag^{-1}$. A high specific capacitance of 297.0 F $g^{-1}$ was attained. After 2000 cycles (2 $Ag^{-1}$), the specific capacitance retained was >90% of the original value. The supercapacitor had an energy density and power density of 15.8 $WhKg^{-1}$ and 4000 $WKg^{-1}$, respectively. Yang et al. [111] developed a supercapacitor electrode based on the PMMA and cobalt-doped hollow porous carbon nanofiber-derived foam. The supercapacitor electrode had a high specific capacitance of 446 $Fg^{-1}$ at 0.5 $Ag^{-1}$. The electrode had long-term stability and catalytic activity of 0.842 V, i.e., analogous to a commercial Pt/C catalyst. Thus, the PMMA nanocomposite foams have been acknowledged as promising materials in energy storage and conversion.

## 6. Challenges and Summary

In this review, considering PMMA as a foam matrix, PMMA/CNT, PMMA/graphene, PMMA/graphene oxide, PMMA/nanoclay, PMMA/silica, PMMA/POSS, and other nanocomposite foams have been presented (Table 3).

**Table 3.** Specifications of PMMA nanocomposite foams.

| Matrix | Nanofiller | Foaming | Property/Application | Ref |
|---|---|---|---|---|
| PMMA | CNT | Supercritical $CO_2$ foaming | Young's modulus; collapse strength | [52] |
| PMMA | CNT | Supercritical $CO_2$ foaming | Cell size 3.0–3.8 μm, cell density $1.8 \times 10^{10}$–$3.7 \times 10^1$ Cells/$cm^3$ | [53] |
| PMMA | CNT | Supercritical $CO_2$ foaming | Cell size; cell density; tensile strength; tensile modulus | [54] |
| PMMA | CNT | Supercritical $CO_2$ foaming | Electrical conductivity $3.34 \times 10^{-6}$–$4.16 \times 10^{-6}$ $Scm^{-1}$ | [55] |
| PMMA | CNT | Supercritical $CO_2$ foaming | Increased cell density; reduced cell size | [56] |
| PMMA | Reduced graphene oxide | Supercritical $CO_2$ | Electrical conductivity 0.160–0.859 $Sm^{-1}$; microhardness 303.6–462.5 MPa; thermal conductivity 0.35–0.70 W/mK | [62] |
| PMMA | Graphene oxide | Solution blending; supercritical $CO_2$ | Morphology; cell density; cell size | [65] |
| PMMA | Montmorillonite | Subcritical $CO_2$ | Tensile strength 12.49 MPa | [69] |
| PMMA | Organically modified montmorillonite | Supercritical $CO_2$ | Elastic modulus 2959–3449 MPa·$cm^3$·$g^{-1}$; Glass transition temperature 140–142 °C | [70] |
| PMMA | Cloisite | Gas foaming | Maximum decomposition temperature 388 °C | [71] |
| Poly(methyl methacrylate)/polyurethane | Cloisite | Mixing/heating | Thermal stability | [73] |
| PMMA | POSS | Co-rotating twin-screw extruder | Thermal conductivity 0.16 W/mK; $T_g$ 119 °C; thermal degradation temperature 375 °C | [78] |
| PMMA | Silica | Supercritical $CO_2$ foaming | Thermal conductivity 0.072 $Wm^{-1}K^{-1}$ | [79] |
| PMMA | Silica | Supercritical $CO_2$ foaming | Cell density $7.5 \times 10^8$–$4.8 \times 10^{11}$ | [80] |
| PMMA | Silica | Supercritical $CO_2$ foaming | Compressive strength 18.90–18.12 MPa | [81] |

The properties of PMMA nanocomposite foams have been tailored with low nanofiller contents. Both nanocarbon and inorganic nanofillers have been employed as significant nanofillers for the PMMA foam matrix. The nanocarbons and inorganic nanofillers appear to be the cutting-edge material to raise the electrical conductivity, mechanical constancy,

thermal stability, and physical properties of the nanocomposites. Moreover, the reduced cell size and increased cell density resulted from nanofiller inclusion. The nanofillers have been supposed to form an interconnecting network in the foam matrix to reduce the cell size and increase cell density. Advancements in PMMA-based nanocomposite foams have been studied for the important applications to date. Various facile techniques such as foaming, solution mixing, etc. have been used to form PMMA nanocomposites. PMMA nanocomposites have been developed using an in situ polymerization and solution method. The foaming methods employed include supercritical $CO_2$, the high pressure method, the melt method, and the use of foaming agents. High-performance PMMA nanocomposite foams have been produced through effective dispersion during in situ polymerization of MMA monomer and subsequent foaming. In situ polymerization has been frequently used to convert the MMA monomer to PMMA. This process involves the use of an initiator such as azo-bis-isobutyronitrile. Heating up to 55 °C for 24 h is also recommended to completely convert the MMA to PMMA. The interaction between the PMMA–nanofiller, nanocellular foam structure, and homogeneity of the cells offered substantial improvement in nanocomposite foam properties. In this regard, appropriate nanofiller functionalization may enhance dispersion and interaction with the matrix. Due to better nanofiller dispersion, PMMA nanocomposite foams have shown substantial improvement in their physical features. An important use of PMMA nanocomposite foams has been experiential for EMI shielding devices. PMMA reinforced with CNT and graphene has been foamed using subcritical $CO_2$ to form the EMI shielding material [53,96,97]. High performance EMI shielding devices can be produced due to the inclusion of nanoparticles in the PMMA foam matrix. Functional nanofillers could be applied to future EMI shielding devices to attain high shielding efficiency. Efforts have also been experiential regarding PMMA nanocomposite foam-based sensors [102,103]. However, very limited designs have been explored for gamma ray sensors so far. Future research is needed to expand research on PMMA nanocomposite foam-based sensors with higher detection limits that have electronic, chemical, and biological sensing features. Moreover, the use of functional nanoparticles may enhance the prospect of sensors for advanced future designs. Successful attempts have been observed for PMMA nanocomposite foam-based supercapacitors, relative to traditional energy storage devices [110,111]. However, comprehensive efforts are needed to produce more supercapacitor designs and expand this field. Even now, there are several unaddressed technical areas (automotive, electronics, biomedical, etc.) which could be enhanced using PMMA nanocomposite foams. Understanding the mechanism of physical or chemical interactions in PMMA nanocomposite foam is essential for future progress. Moreover, the use of novel functional nanofillers and innovative fabrication techniques need to be developed.

This review summarizes the essential aspects of PMMA nanocomposite foams. Various facile strategies have been used to design these nanocomposites. Initially, the fundamentals of PMMA and PMMA foams have been discussed. Later, the PMMA nanocomposite foams have been prepared with CNT, graphene, nanoclay, silica, POSS, and other nanoparticles. The superior thermal stability, $T_g$, mechanical strength, compressive strength, electrical conductivity, thermal conductivity, etc. of the nanocomposite foams have been summarized. Consequently, PMMA nanocomposite foams have been explored for radiation shielding, supercapacitor, and sensor applications. This review will be pioneering in the field of PMMA nanocomposite foams to expand future research in this area.

**Author Contributions:** Conceptualization, A.K.; data curation, A.K.; writing of original draft preparation, A.K.; Review and editing, A.K.; P.B. All authors have read and agreed to the published version of the manuscript.

**Funding:** This research received no external funding. This is an invited article with waived off charges.

**Institutional Review Board Statement:** Not applicable.

**Informed Consent Statement:** Not applicable.

**Data Availability Statement:** Not applicable.

**Conflicts of Interest:** The authors declare no conflict of interest.

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
