# Peer review of "Poly(methyl methacrylate) Nanocomposite Foams Reinforced with Carbon and Inorganic Nanoparticles—State-of-the-Art"

_jcs, doi:10.3390/jcs6050129_

Round 1

Reviewer 1 Report

The manuscript entitled “Poly(methyl methacrylate) Nanocomposite Foams Reinforced with Carbon Nanotube, Graphene, Nanoclay and POSS Nanoparticles — State-of-the-art “ by Ayesha Kausar  and Patrizia Bocchetta reports the progress recorded in the case of the nanocomposites based on poly(methyl metacrylate), nanocarbon nanofillers (carbon nanotube, graphene) and inorganic nanoparticles (nanoclay and polyhedral oligomeric silsequinoxane, silica, etc.). The discussions are well structured. However, in the revised manuscript, it is necessary :

i) to explain the notation PMMA/CNT 0.5 and PMMA/CNT 1 from Table 2 ;

ii) to clarify the CNT type (SWNT, DSWNT, MWNT) used in References 51- 57, 91, 92, 94-96  as well as in Table 3;

iii) to show the details about the in situ polymerization of PMMA-based nanocomposites ;

iv) a copyright for all figures is necessary to be obtained and mentioned in the main text.

I recommend this article to be published in the Journal of Composite Science only a minor revision.

Author Response

Reviewer 1:

  1. Table 2, The terms are defined n caption (highlighted).
  2. The terms CNT is used in some original studies and CNT type is not mentioned. So, it’s better to present as it is. CNT type is given where necessary.
  3. Line 393-397, The details on in situ polymerization of PMMA are given.
  4. Permissions for all the adopted figures have been obtained using Rightslink online copyright clearance system. Now mentioned in Fig/Table captions.

Reviewer 2 Report

The manuscript is a critical review of the data available in the literature, which addresses important and relevant issues related to the synthesis and investigation of the properties of PMMA-based nanocomposite foams. This is the first review in the field of nanocomposite polymer foams. Various strategies for the synthesis of PMMA-based foams and PMMA- nanocomposites-based foams including inorganic nanofillers such as carbon nanotubes, graphene, nanoclay, polyhedral oligomeric silsesquioxane, silica, etc. are described. The properties of the foams obtained, such as thermal stability, electrical and thermal conductivity, excellent mechanical properties, especially compressive strength, are discussed. Of particular interest is the section describing the prospects for using nanocomposite foams as electromagnetic shieldings, sensors and electrode materials for electrochemical energy storage systems. The review analyzes a rather large amount of information, the list of references includes 111 titles.                      I recommend accepting current manuscript, however, there are some aspects in this manuscript that should be improved:

  1. Lines 25 and 299 are typos.
  2.  Line 220 - A sentence beginning with the word Silica should open a new paragraph
  3.  Line 237 - typo, should be (EMI shielding)
  4.  Line 240 - carbon nanotube should be changed to CNT
  5.  Line 242 - electromagnetic interference shielding should be changed to EMI shielding
  6.  Line 248 - error: percolation threshold cannot be expressed in conductivity units
  7.  Line 273 - error: conductivity is expressed in units of Sm.cm-1
  8.  Line 296 - supercapacitor, ultracapacitor and electrochemical capacitor are the same thing
  9.  Line 338 - "nanocomposite" is repeated twice
  10.  You should not use the acronym (POSS) in the title of the article. This abbreviation is also used in the conclusion. However, it is not deciphered in the text of the article.
  11.  The title of the article is very cumbersome. Not all used nanoparticles are listed (for example, silica). It is advisable to change the title of the article, for example: “Poly(methyl methacrylate) Nanocomposite Foams Reinforced with Carbon and Inorganic Nanoparticles—State-of-the-art”

Author Response

Reviewer 2:

  1. Line 29, corrected. Line 299, Ok.
  2. Line 260, “Silca ……” new para now.

3.Line 279, EMI shielding, corrected.

  1. Line 197, 281, CNT corrected.
  2. Line 283, EMI corrected.
  3. Line 289, The units of percolation threshold removed.
  4. Conductivity units are already correct, according to the literature.
  5. Line 347, 348, Yes, that’s why word “or” is used between the terms.
  6. Line 400, 329, 330, nanocomposite is revised.
  7. Line 251, 252, Polyhedral oligomeric silsesquioxane (POSS) already defined in text. In title full name is given.
  8. Title is revised, as suggested.

Round 2

Reviewer 1 Report

I propose to be published this manuscript in Journal Composite Science in the present form.  

Reviewer 2 Report

The authors took into account all the comments of the reviewer. The reviewer accepted the authors' answers and agrees with most of the included corrections to the text. The reviewer is not completely satisfied with the responses to comments 6 and 8 and suggestss the following revisions:

Lines 249-250 - For a better understanding of the text, I suggest you edit the text as follows: The SWCNT loading enhanced the percolation threshold. At a PMMA/SWCNT percolation threshold of 3 weight %, the increase in electrical conductivity is 13 orders of magnitude from 10-15 to 10-2 Scm-1.

Lines 349-350 - The phrase is incorrect. Not only supercapacitors are effective energy storage devices, but also batteries. They differ in the mechanism of energy storage. Since only supercapacitors are discussed in the review, I suggest that this phrase be written as follows: "Supercapacitors are one of the most efficient energy storage devices". As stated in the previous review, super-, ultra-, and electrochemical capacitors are one and the same.

Line 252 - Missing letter